# Learning from Monocyte-Macrophage Fusion and Multinucleation: Potential Therapeutic Targets for Osteoporosis and Rheumatoid Arthritis

**DOI:** 10.3390/ijms21176001

**Published:** 2020-08-20

**Authors:** Laura Gambari, Francesco Grassi, Livia Roseti, Brunella Grigolo, Giovanna Desando

**Affiliations:** IRCCS Istituto Ortopedico Rizzoli, Laboratorio RAMSES, 40136 Bologna, Italy; laura.gambari@ior.it (L.G.); francesco.grassi@ior.it (F.G.); giovanna.desando@ior.it (G.D.)

**Keywords:** bone loss, osteoporosis, rheumatoid arthritis, macrophage fusion and multinucleation, osteoclasts, giant cells, inflammation, macrophage polarisation, natural compounds

## Abstract

Excessive bone resorption by osteoclasts (OCs) covers an essential role in developing bone diseases, such as osteoporosis (OP) and rheumatoid arthritis (RA). Monocytes or macrophages fusion and multinucleation (M-FM) are key processes for generating multinucleated mature cells with essential roles in bone remodelling. Depending on the phenotypic heterogeneity of monocyte/macrophage precursors and the extracellular milieu, two distinct morphological and functional cell types can arise mature OCs and giant cells (GCs). Despite their biological relevance in several physiological and pathological responses, many gaps exist in our understanding of their formation and role in bone, including the molecular determinants of cell fusion and multinucleation. Here, we outline fusogenic molecules during M-FM involved in OCs and GCs formation in healthy conditions and during OP and RA. Moreover, we discuss the impact of the inflammatory milieu on modulating macrophages phenotype and their differentiation towards mature cells. Methodological approach envisaged searches on Scopus, Web of Science Core Collection, and EMBASE databases to select relevant studies on M-FM, osteoclastogenesis, inflammation, OP, and RA. This review intends to give a state-of-the-art description of mechanisms beyond osteoclastogenesis and M-FM, with a focus on OP and RA, and to highlight potential biological therapeutic targets to prevent extreme bone loss.

## 1. Introduction

Bone diseases, such as osteoporosis (OP) and rheumatoid arthritis (RA), are an enormous burden for the healthcare system worldwide, mainly due to the enhanced risk for bone fractures [1]. Both diseases display excessive bone resorption by osteoclasts (OCs), leading to bone destruction. In OP, the bone loss depends on the impaired bone remodelling. Uncoupling between bone formation supported by osteoblasts (OBs) and bone resorption by OCs in favour of resorption activity is one of the main pathognomonic mechanisms in OP [2]. In RA, the hyperproduction of inflammatory cytokines and matrix-degrading enzymes from activated immune cells in the synovial membrane contributes to driving joint destruction, including subchondral bone loss [3]. Besides their role in the immune system, many inflammatory cytokines modulate OCs recruitment and differentiation and OBs activity, leading to lower bone formation at sites of bone erosion [4,5]. Beyond OCs, emerging cell players are multinucleated giant cells (GCs) [6]. Despite their different functions, OCs and GCs share a common origin because they derive from the differentiation and fusion of monocyte-macrophage lineage progenitors [7].

Interestingly, monocytes and macrophages exhibit a pronounced fusogenic potential. Depending on the anatomical site and environmental milieu, they can create two specific cell types: mature OCs in bone and GCs as part of the immune response [7]. A typical characteristic of OCs and GCs is multinucleation, an essential step for promoting their maturation [8,9]. Defective multinucleation of OCs and GCs leads, respectively, to impaired bone resorption [9] and increased susceptibility to chronic inflammatory diseases [6].

In general, OCs regulate bone homeostasis in the entire life course during skeletal growth and development and bone repair following tissue injuries [10,11]. GCs instead enhance tissue-specific phagocytic activity when macrophages are not sufficient [12]. In pathological conditions related to inflammation, GCs produce specific signals, which can stimulate monocyte subset to differentiate into OCs [13,14,15]. RA patients show GCs distribution not only in the subchondral bone tissue but also in the cartilage and synovial membrane, and their number correlates with synovitis severity and enhanced OCs numbers in the bone [15].

Altogether, M-FM stands at the interface between physiological and pathological responses because it is modulated by several cells and molecular signalling pathways, which are still far to be elucidated. Gathering a better grasp of cellular and molecular mechanisms involved in M-FM can offer valuable prospects on potential biological targets for treating OP and RA. In this review, we intend to present an overview of how several modulators influence M-FM during bone matrix turnover and inflammatory conditions by highlighting the gaps remaining in the literature. Finally, we discuss challenges and prospects to improve therapeutic alternatives for OP and RA.

## 2. M-FM during Normal Osteoclastogenesis: Therapeutic Perspectives for OP

### 2.1. Morphological Features of OCs in Physiological Conditions

OCs are bone-resorbing cells which can arise from immature monocytes and mature tissue macrophages [16]. Immature cells from the monocyte-macrophage lineage upon macrophage colony-stimulating factor (M-CSF) and receptor activator for nuclear factor κ B ligand (RANKL) differentiate into OCs. Mature OCs are multinucleated (2–20 nuclei) cells (up to 100 µm) with a polarised conformation. Not all the nuclei of OCs are transcriptionally active in each stage of differentiation. Nuclear factor of activated T-cell cytoplasmic 1 (NFATc1) is a master transcription factor for OCs differentiation, present in most nuclei only in early differentiated OCs and at a less extent in further stages [17]. OCs show a large cytoplasm volume per each nucleus and enclose many vacuoles, mitochondria and lysosomes. OCs surface membrane displays four domains: the sealing zone (SZ), the ruffled border (RB), the basolateral domain (BD) and the functional secretory domain (FSD) [18]. SZ is a dynamic actin-rich structure that keeps the boundary for the resorption area of the bone [19]. RB also called as specialised lysosome-related organelle (LRO), functions as the OCs’ secretory apparatus for protons (H^+^), chloride ions (Cl^−^), and proteases during bone degradation [18,20]. BD is the site where endocytosis occurs [21], whereas FSD regulates the trafficking and secretion of vesicles [18]. Cell-cell fusion requires cytoskeleton rearrangement and assembly, cell polarisation and multinucleation of competent cells. In particular, the size and the number of multi-nuclei of OCs depend on the status of the actin cytoskeleton signalling [22]. Several authors described a direct relationship between OCs size and resorption activity. A 10-fold increase in cell radius results in a 10-fold increase in resorption areas because of more energy (ATP) and release of proteases [23]. The large OCs size is attained either through monocyte fusion, or through fusion-independent cytoplasm growth, or with a combination of these processes. Several studies identified a repertoire of actin-rich structures in OCs crucial to the fusion, including circumferential podosomes [24], zipper-like structures (ZLS) [22,25], and tunnelling nanotubes (TNT) [26]. In particular, the formation of ZLS is vital to create a discontinuous broad contact surface between the two apposed plasma membranes of two fusion partners and to produce efficient M-FM; its absence contributes to the formation of smaller OCs [25,26,27].

### 2.2. Osteoclastogenesis and Osteoclastic Bone Resorption

Osteoclastogenesis is a multi-step process, controlled via the spatiotemporal regulation of several differentiation factors, driving OCs maturation and commitment towards bone resorption (Figure 1). Osteoclastogenesis and OCs resorption are energy-consuming processes, bolstered by the mitochondrial oxidative metabolism and the glycolysis [28]. During the initial phases, a gradient of chemokines recruits mononuclear hematopoietic precursors [29]. Macrophage colony-stimulating factor (M-CSF) and receptor activator for nuclear factor κ B ligand (RANKL) activate the OCs differentiation from pre-OCs by binding with their receptors c-fms and RANK. M-CSF is pivotal for stimulating the proliferation and the first stages of OCs maturation. RANK/RANKL and the decoy receptor osteoprotegerin (OPG) are central regulators of OCs development and function [30,31]. The rate of RANKL to OPG in the bone serves as an index of the net stimulus for osteoclastogenesis. Molecular cross-talk between RANK-RANKL determines cytoplasmic Ca^2+^ oscillations and triggers tumour necrosis factor (TNF) receptor-associated factor (TRAF) family proteins such as TRAF6. RANK-RANKL signalling activates several downstream pathways like nuclear factor-kappa-B (NF-kB), c-Jun N-terminal kinases (JNK) and c-Fos, mitogen-activated protein kinase (MAPK), and protein kinase B (AKT), phosphatidylinositol-3 kinase (PI3K). Activator protein-1 (AP-1), NF-𝜅B, NFATc1, tartrate-resistant acid phosphatase (TRAP), cathepsin K (CTSK) and calcitonin receptor (CTR) are among the central genes needed for OCs development [32]. Osteoclastogenesis also requires co-stimulatory signalling activated by the binding of immunoreceptors tyrosine-based activation motif (ITAM), bearing the adaptors DNAX-activating protein 12 (DAP12) and Fc receptor standard g chain (FcRγ), with other immunoreceptors. DAP12 form a complex with Siglec-15 or TREM2 receptors; FcRγ pairs with OCs-specific activating receptor (OSCAR), immunoglobulin receptor A (PIR-A), and Fc receptors. RANKL-NFATc1 signalling also activates B lymphocyte-induced maturation protein 1 (Blimp1), which acts as a transcriptional repressor of anti-osteoclastogenic genes such as Irf8, Mafb, BCL6 [33] and induces various OCs genes like dendritic cell-specific transmembrane protein (DC-STAMP), ATPase H1 transporting V0 subunit d isoform 2 (ATP6V0d2), TRAP, CTSK, and NFATc1 [34]. miRNAs are small non-coding RNA, which regulate several genes at the post-transcriptional level, including those involved in OCs differentiation and fusion [35,36,37]. As a consequence, their levels of expression are strictly modulated during osteoclastogenesis. The redox balance exerts a vital role in regulating osteoclastogenesis. RANKL induces a transient and fast increase in reactive oxygen species (ROS) through activation of TRAF6, NOX1, RAC1 [31,38], which is critical for OCs differentiation. Finally, key signalling modulating osteoclastogenesis derives from cell-cell contacts mediated by proteins such as ephrins. Interestingly, cell-cell communication between OCs and OBs through ephrinA2 (expressed in OCs downstream to RANKL signalling)-EphA2 (expressed in OBs) activates the initiation phase of bone remodelling by enhancing OCs differentiation and suppressing OBs differentiation [39]. Conversely, cell-cell contact between OCs and OBs through ephrinB2 (expressed in mature OCs)-EphB4 (expressed in OBs precursors) enhances osteogenic differentiation and suppresses OCs function [39].

M-FM is the last process in OCs differentiation, critical for ensuring optimal OCs function through cytoskeleton reorganisation and formation of the RB and SZ. After M-FM and the SZ formation, OCs can adhere to the bone via αβ1 integrin. The SZ maintains local acidification thanks to the ion channels: Clcn7 (which encodes chloride Channel 7), and ATP6v0d2 [40,41]. During OCs resorption, RF releases LRO containing matrix-degrading metalloproteinases (MMPs); TRAP and CTSK [40,41].

### 2.3. Regulators Modulating M-FM

The concept of M-FM as the result of a random cell-cell fusion between mononuclear pre-OCs has evolved [42]. Recent experimental evidence suggests that the fusion process occurs between specific “fusion partners” whose selection depends on the nuclear number, differentiation and mobility level [43]. Differential expression of DC-STAMP, CD47, CxCr3, syncytin-1 in cells at varied maturation stages might drive this “selection” [44]. This autonomous system could prevent pre-OCs fusion in the bone marrow and regulate their size and nuclei number [44]. CD44/MMP-9 complex is a unique motility-enhancing signal [45], critical for both migration and cell fusion [46]. CD44 is a component of the podosome, which binds to hyaluronic acid, collagen, osteopontin, and laminin [47]. CD44 assembles MT1-MMP to the cell surface and regulates MMP-9 activity; thus promoting OCs migration [48]. The chemokine monocyte chemoattractant protein-1 (MCP-1) and its receptor CCR2 have a compelling role in fusion. CCR2 knockout mice showed OCs with smaller size and number of nuclei and overall, a lower number of OCs [49]. MCP-1-deficient mice showed limited multinuclear OCs formation and expression of DC-STAMP, NFATc1, and CTSK [50]. Pre-OCs attachment to the bone matrix launches cell migration and fusion. CD9, DC-STAMP, OC-stimulatory transmembrane protein (OC-STAMP) are among the main modulators involved in cell fusion, regulated by RANKL. OC-STAMP and DC-STAMP are unique OCs-specific fusogenic molecules, whereas CD-9 is a permissive fusogen, not only restricted to OCs [24,44,51]. After stimulation with RANKL, OC-STAMP mRNA increases with time, culminates at 48 hrs and later decreases [44]. NFATc1 binds to the promoter of DC-STAMP and increases its expression in pre-OCs [52]. Surface distribution of fusogenic molecules increases in pre-OCs before cell fusion and declines after the fusion [44,51,53,54,55]. Depletion of CD-9, DC-STAMP, and OC-STAMP suppressed formation of TRAP multinuclear OCs (>10 nuclei/cell) in various studies, leading to inhibition of OCs resorption [44,51,56,57]. Syncitin-1 is another crucial protein, which upholds the fusion of multinucleated cells but not of mononucleated cells [58]. Among membrane proteins, the sialic acid-binding immunoglobulin-type lectin 15 (Siglec-15) rises after RANKL stimulation by NFATc1 [59,60]. Siglec-15 functions as a coupling receptor in the co-stimulatory ITAM signalling by cooperating with the adaptor DAP12 [61] and activates different downstream pathways. In specific, it strengthens the phosphorylation of Erk and PI3K/Akt, downstream of RANK–TRAF6 pathway; while the pathway involved in TNF-α stimulation remains uncertain. Siglec-15 is specifically involved in M-FM, given that Siglec-15^−/−^ mice display a lower number of multinucleated OCs [59,62] and inhibit OCs activity [63]. Another process critically related to cell fusion is lysosomes trafficking. Rab27a localised in lysosomes increases during OCs differentiation from macrophages and mediates membrane trafficking events. Rab-27a deficient OCs display abnormal lysosomal protein distribution and impaired bone resorption [64]. Osteoclastogenesis-associated transmembrane protein-1 (Ostm1) inhibits OCs fusion by restraining the NFATc1 pathway through the modulation of calcium signalling response [65].

Interestingly, a few miRNAs would seem specifically implicated in OCs fusion. miR7b and miR30a directly target the mRNA of DC-STAMP [37,66]; miR-26a increases the expression of DC-STAMP and ATP6v0d2 by directly targeting connective tissue growth factor/CCN family 2 (CTGF/CCN2) [66]; miRNA124 targets Rab27a [67]. Despite the low knowledge about their control and mechanism of action, they create a fusion-competent status and stimulate fusion between the two lipid bilayer membranes, which do not spontaneously undergo fusion [24]. Several other molecules have been linked to OCs fusion [55,58,68,69,70,71,72,73,74,75,76,77,78,79,80,81,82,83,84,85] and listed in Table 1. Moreover, a specific plasma membrane structure is necessary to cell-fusion: the lipid raft domain [51]. Disruption of this domain causes the complete inhibition of the formation of multinuclear OCs in the presence of RANKL [51].

### 2.4. OCs and Osteoclastogenesis during OP

OP is one of the most severe bone disorders worldwide, affecting about 200 million patients [1], mainly in post-menopausal women and older people. The key feature of OP is a deterioration of the bone composition and subsequent bone fragility, leading to a severe risk of bone fractures [86]. OP has a multifactorial aetiology. Post-menopausal and senile forms of OP are the most common. They are caused by oestrogen loss and calcium deficiency with a critical role of the immune system in their physiopathology [87]. The secondary types of OP, which are triggered by diseases (such as RA, diabetes mellitus), use of drugs (such as glucocorticoids) and improper inhabits (such as nutritional deficiency, smoke etc.), and idiopathic juvenile OP are less widespread [88]. The uncoupling of bone resorption and bone formation (Figure 2) is the primary pathogenetic process in OP [89,90]. Biological mechanisms involved in this complex phenomenon are several and still partly unknown. Increased osteoclastogenesis and OCs lifespan and decreased osteoblastogenesis and OBs function occur in OP. Changes in local and systemic growth factors or hormones influence bone tissue. In specific, oestrogen contribute to the inhibition of OCs by reducing RANKL on marrow cells and increasing OPG secretion by OBs. Oestrogen increases the thiol antioxidants defences in OCs and suppress TNF-α [91]. During post-menopausal OP, oestrogen decline contributes to increasing RANKL secretion by OBs and osteocytes, which in turn increases bone resorption [92,93].

Interestingly, oestrogen plays an important role in modulating immune responses by controlling the functions of T cells and macrophages with essential implications in bone metabolism [94]. Oestrogen deficiency contributes to increasing inflammatory cytokines leading to increased osteoclastogenesis [87,95]. TNF-α, produced by bone marrow T lymphocytes, is among the main cytokines in the oestrogen deficiency-induced bone loss [87]. In particular, it enhances OCs formation and increases the responsiveness of pre-OCs to RANKL thanks to Nuclear Factor kappa-light-chain-enhancer of activated B cell (NFκB) and AP-1 signalling pathways. Moreover, it blocks osteoblastogenesis by impairing the function of bone-forming OBs [96]. The microbiota is among the factors activating T cells in sex steroid deficiency–associated bone loss during OP. Increased gut permeability contributes to triggering inflammatory signalling pathways [97]. Beyond T cells, B cells and macrophages also contribute to the production of osteoclastogenic cytokines [96,98]. Recently, the relationship between macrophage polarisation and oestrogen is gaining considerable attention among scientists. Several studies showed that oestrogen promote M2 macrophages (with a wound-healing phenotype) and inhibit M1 subset (with a pro-inflammatory phenotype). Oestrogen deficiency is responsible for an increased M1/M2 macrophage ratio, leading to increased production of osteoclastogenic factors. Oestrogen-dependent effects on macrophage subsets may be a potential target for pharmacological approaches in post-menopausal OP [99].

Oxidative stress also has a vital role in the pathophysiology of OP, leading to the production of ROS. ROS formation promotes the apoptosis of OBs and osteocytes in favour of osteoclastogenesis [100] and takes part also in OCs differentiation [100]. Excessive ROS production is among the most frequent pathological aspects of the skeletal involution following ageing and loss of sex steroids [101,102].

M-FM participates in the pathogenesis of a vast range of disorders; several inflammatory osteolysis diseases show excessive OC fusion and bone resorption [103]. It is still uncertain whether the alteration of M-FM is a phenotypical feature of OP. OCs in the bone resorption lesions of OP display elevated levels of CD9, a permissive fusogen [51]. Oestrogen concentration is essential in regulating some OCs-specific fusogenic molecules. A high concentration of oestrogen downregulates the expression of OC-STAMP, thus suggesting that oestrogen depletion during post-menopausal OP can increase OC-STAMP levels and multinucleation [104].

### 2.5. Therapeutic Strategies in OP

OCs are the primary target of bone sparing therapies due to their central role in physiologic bone development, remodelling and function. A better grasp of their intracellular signalling pathways can be helpful to find novel therapeutic targets. In the past, oestrogen replacement (ER) was the most used therapy, given that oestrogens directly contribute to OCs inhibition [105,106]. Although ER and selective oestrogen receptor modulators reduce the incidence of OP-related fractures, their use generates several side effects [107,108]. Along this line, experts have examined several alternatives to both neutralise bone destruction and favour anabolic processes (Figure 3). Nowadays, the first-line regimens for OP are bisphosphonates (BPs), which are anti-resorptive treatments. BPs bind avidly to hydroxyapatite crystals on bone surfaces [109,110] and are up-taken by OCs during the bone resorption [111]. BPs are divided into two classes with different mechanisms of action. Non-amino-BPs cause OCs apoptosis. Amino-BPs cause changes in OCs cytoskeleton, leading to impaired OCs function [112]. Despite their excellent results in reducing risk fractures [113], BPs cannot rebuild the injured bone architecture [114] and show several side effects. Long-term use of BPs compromises bone strength because of (i) unintended inhibition of coupled bone formation; (ii) impairment in the bone remodelling; and (iii) decrease in the OBs number and bone formation rate [115]; thereby, contributing to low bone repair ability [116,117,118]. Several efforts in finding new classes of anti-resorptive treatments are ongoing. Among the most promising alternatives, Denosumab, a monoclonal antibody to RANKL, blocks OCs maturation, function and survival, thus reducing bone resorption and risk fractures [119,120,121]. It is not incorporated into the bone matrix-like BPs, and bone turnover is not suppressed after its cessation; thus, continuous administration sustains its biological activity [122]. Using anabolic agents has gained growing interest among scientists for their ability to promote bone formation through the activation of OBs. Treatments with parathyroid hormone (PTH), PTH analogues and sclerostin inhibitors (such as the sclerostin monoclonal antibody romosozumab) are regimen treatments to encourage bone anabolism [123,124,125]. They effectively reverse bone damages in OP by restoring the structure of trabecular and cortical bone. However, these anabolic treatments are used for a limited time due to several side effects [126] and their incapacity to prevent bone resorption; thus, patients should always receive anti-resorptive agents to increase bone mineral density gain [127]. Identifying novel classes of remedies capable of “uncoupling” bone resorption from the bone formation by favouring the latter is highly demanding. In this line, strontium ranelate inhibits OCs function while promoting OBs proliferation, by uncoupling bone formation and bone resorption but shows several side effects [128]. Selective inhibitors of osteoclastic hydrogen ion transport and CTSK are among the newer techniques for inhibiting bone resorption [129,130,131]. They remove active OCs without impairing OCs differentiation, which is crucial to ensure proper OCs-OBs coupling during bone remodelling. While CTSK inhibitors present several safety concerns, V-ATPase efficacy and safety remain undefined, [131].

#### Novel Targets for Anti-Osteoclastogenic Therapies for OP

Targeting both bone resorption and M-FM selectively could represent a strategic alternative to reduce OCs functions while preserving OBs-OCs communication. In this light, therapeutic options based on natural compounds could act on specific phases of OCs formation like fusion and multinucleation. In the past few years, different families of nutraceuticals showed inhibitory effects on OCs differentiation and bone resorption [132,133]. Although they inhibit critical genes related to OCs cell-cell fusion (DC-STAMP, ATP6v0d2, OC-STAMP) and alter the creation of the actin ring, they do not inhibit in a specific manner M-FM. Besides natural compounds [133], other molecules like miRNA can specifically inhibit M-FM and fusion. In specific, miR7b, miR30a target the unique OCs-specific fusogen DC-STAMP [37,134]; miR26a targets CTGF/CCN2 and inhibits DC-STAMP [66]; miR124 targets Rab27a [67]. Inhibiting DC-STAMP can provide both decreased OCs activity and enhanced bone production by OBs, leading to increased bone mass [117]. The overexpression of miR7b, miR26a, miR30a in pre-OCs can inhibit OCs multinucleation, actin-ring formation, and bone resorption [37,66,135]; therefore suggesting their potential use for inhibiting M-FM in vivo. Recently, a polyethylenimine (PEI) functionalised graphene oxide complex loaded with miR-7b overexpression plasmid has been tested in ovariectomised (OVX) mice. Herein, miR-7b overexpression abrogated OCs fusion and bone resorption while maintaining mononuclear preOCs [134]. This research provides preliminary evidence on the in vivo potential of specific miRNAs to target M-FM in OP. Developing anti–DC-STAMP and anti–OC-STAMP antibodies is another attempt to block OCs fusogenic function with encouraging in vitro results in suppressing OCs multinucleation [44,136]. To date, no indications are available whether they can modulate OP in vivo.

Blocking Siglec-15 and Rab27a offers further promising opportunities to prevent bone loss and increase bone mass. Blockage of Siglec-15 is highly specific, given that its expression is highly OCs specific. Siglec-15 neutralising antibody induces a rapid internalization of Siglec-15 and inhibits in vitro OCs differentiation, and M-FM in mouse and human bone marrow monocyte/macrophage (BMM) cells stimulated with RANKL [60]. Siglec-15 antibody induce increased bone mineral density in young mice [59,63] and protected against glucocorticoid-induced OP of growing skeleton in juvenile rats [137]. Furthermore, it has been proposed as an effective treatment for juvenile OP [60], as it increases bone mass without any adverse effects on skeletal growth [138]. The unique property of the anti-Siglec-15 Ab of inhibiting M-FM in secondary but not in primary spongiosa can probably depend on a collagen II-OSCAR-dependent compensatory signalling for Siglec-15 inhibition in the primary but not in the secondary spongiosa [138]. miR-124 targets Rab27a and inhibits M-FM, therefore rising its potential use for OP [67]. Although these inhibitors of multinucleation hold good promises as therapeutics for OP (Table 2), clinical trials are mandatory to validate their efficacy, safety and their potential superiority to current drugs.

## 3. GCs and OCs in RA: Relationships with Bone Erosion and Therapeutic Alternatives

### 3.1. Leading Characteristics of GCs: Similarities and Differences with OCs

GCs are multinucleated polykarions macrophages (up to 100–200 nuclei) that likely OCs originate from monocyte-macrophage lineage and cover essential roles in a variety of processes, including foreign body reactions, infections and inflammatory disorders [14,139]. GCs formation seems to be an adaptation process for enhancing phagocytic activity when macrophages cannot degrade large biomaterials or tissue irritants [12,140]. Insults during RA like dysregulated immunity and cytokine network and enhanced chondrocyte/OCs activation contribute to inducing macrophage fusion and GCs formation [141]. Depending on the fusion area, organelles arrangement and cytokines, three classes of GCs, with distinct histological features, can form Langhans giant cells (LGCs), Touton giant cells (TGCs), and foreign body giant cells (FBGCs) [7,14]. In particular, Interferon gamma (IFN-γ) and IL-13 induce LGCs, which display nuclei surrounding the Golgi apparatus and other organelles. They mainly exhibit an inflammatory phenotype with implications during granulomas and RA [139]. M-CSF, Interleukin 6 (IL-6), and IFN-γ promote TGCs in many pathologic processes, such as xanthomas and granulomas [14]. IL-4 and IL-13 stimulate FBGCs, which counteract the inflammatory responses [142]. GCs, unlike OCs, cannot resorb bone and express several M2 markers like Ym1, Fizz1, CD-206, arginase-1 and arachidonate 15-lipoxygenase (Alox 15), an enzyme required in wound-healing and termination of inflammation [143]. Overall, GCs are found in several anatomical sites such as cartilage and the synovial membrane in RA patients. Authors showed a strong correlation between GCs and synovitis severity together with an enhanced OCs number in the bone [15].

Differently from OCs, GCs cannot create lacunar pits of resorption in the bone [15]; they adhere to the bone and reduces the mineral phase without digesting the matrix fraction [7]. GCs express CD-11c, CD-68, HLA-DR, and DC-sign like dendritic cells and can select lymphocyte co-stimulatory molecules [15], thus displaying an active role during antigen presentation. GCs and OCs display different integrins for the fusion among cell types. GCs precursors exhibit the integrins αβ2, whereas αβ3 mediates OCs fusion and cytoskeleton rearrangement [7]. GCs express higher levels of CCL2, CCL3, CCL4, CCL5, CCL9, and GM-CSFR when compared to OCs [15]. CCL-2 and its receptor CCR2 are essential mediators which foster the chemotaxis of both OCs and GCs before cell fusion [15]. The knowledge of fusion mechanisms and modulators driving the differentiation of monocyte-macrophage precursors towards OCs and GCs is not well defined [139]. RANKL and the cytokines IL-4 and IL-13 are among the leading cell-type-specific fusion mediators in OCs and GCs [7]. Macrophage fusion receptor (MFR), also known as P84/BIT/SIRPα/SHPS-1, is transiently expressed in macrophages at the onset of fusion by stimulating the differentiation of both OCs and GCs [144]. MFR belongs to the superfamily of immunoglobulins and interacts with CD47 on fusing macrophages [145]. OC-STAMP and DC-STAMP, induced by RANKL-NFATc1 axis, are essential mediators in both OCs and GCs to favour cell fusion. Despite both GCs and OCs that occur following cell fusion in a DC-STAMP-dependent manner, the regulating fusion competency is different. GCs display PU.1 and NF-ĸB mobilisation to the DC-STAMP promoter in GCs, while OCs show c-Fos and NFATc1 recruitment to the DC-STAMP promoter. c-Fos–deficient mice exhibit several GCs along with DC-STAMP activation. In contrast, DC-STAMP^-/-^ mice did not show GCs formation [146]. RANKL and ITAM cooperate by inducing NFATc1 during OCs formation but not in GCs; whereas IL-4 and IL-13 mediate GCs formation but not OCs through the activation of signal transducer and activator of transcription 6 (STAT6) via the binding with E-cadherin [146,147,148]. E-cadherin is also crucial for macrophage fusion and multinucleation for the formation of mature OCs [149]. However, OCs and GCs share several common prefusion mediators, including M-CSF, DAP-12, triggering receptor expressed by myeloid cells 2 (TREM2), purinergic receptor P2X7 (P2RX7), tumour necrosis factor (TNF) and potassium calcium-activated channel subfamily N member 4 (KCNN4) [7]. TNF promotes multinucleation in both OCs and GCs. Its combination with RANKL can act on pre-OCs to promote NFATc1-dependent OCs and M-FM through the activation of c-Jun N-terminal kinase (JNK) pathway. KCNN4 takes part at the onset of fusion, and it is necessary for OCs and GCs formation in rodents and humans, by regulating Ca^2+^ signalling [6]. Monocyte subsets display a distinct pattern of tetraspanins (CD-9, CD81) expression and different capacities to form GCs. The intermediate subset CD14^++^CD16^+^ of peripheral human monocytes fuse faster and produce larger GCs than the other subsets. Although tetraspanins would seem to play an important role in the fusion of intermediate monocytes, the regulation of GCs formation has to be still clarified [150]. Similarly to OCs, miRNAs can regulate macrophage fusion towards GCs. MiR7-a-1 regulates GCs formation by targeting Tm7sf4, which is a fusogenic cell surface [151]. Finally, TRAP, CTSK, and MMP-9, expressed during bone resorption, are OCs specific and detected at low concentrations, or not at all, in GCs [7].

### 3.2. M-FM of OCs and GCs and Osteoclastogenesis during Inflammation

Recently, there has been a growing interest in elucidating the influence of the inflammatory milieu on the differentiation of monocyte-macrophage precursors into multinucleated OCs and GCs [16,152,153]. In particular, macrophages can mainly display two main cell subsets with inflammatory (M1) and alternative wound-healing (M2) phenotypes [154]. During chronic inflammation like RA, macrophages display mainly a pro-inflammatory M1 phenotype (markers CD-80, CD-86, TRL2, TRL4, i-NOS) and release inflammatory cytokines such as TNF-α, IL-1β, IL-6, IL-15, IL-17, IL-23, and IL-34. During wound-healing and repair processes, macrophages show an alternative M2 phenotype (markers CD-206, CD-209, CD-163, FIZZ1, and Ym1/2) and release anti-inflammatory cytokines such as IL-10, IL-1R antagonist, vascular endothelial growth factor (VEGF), IL-33. M2 activation status can, in turn, display heterogeneous and distinct macrophages subtypes: M2a, M2b, M2c, M2d depending on the stimuli [155]. Miyamoto T et al. showed that OCs exhibit similar behaviour of M1 macrophages under inflammatory conditions, whereas GCs display mainly an M2-like phenotype [50]. M1 macrophages would seem the leading players which modulate fusion events of pre-OC towards mature OCs by up-regulating fusogenic molecules such as CD-40, DC-STAMP, E-cadherin, MFR-CD47, and MR [144]. TRAF6 would seem to modulate the polarisation of M1 OCs and M2 GCs during inflammation, thus opening valuable insights for future therapeutic applications [146,156]. Signal transducer and activator of transcription (STAT) signalling pathway plays a pivotal role in modulating macrophage polarization. Among STAT family molecules, STAT-6 contributes to favouring the formation of FBGCs through the promotion of macrophage cell-cell fusion. Conversely, STAT-1 is an inhibitor of FBGCs multinucleation [148]. In general, the STAT-1/STAT-6 axis modulates MF via the regulation of OC-STAMP and DC-STAMP and modulates fusogenic mechanisms in FBGCs [7] (Figure 4). The anti-inflammatory cytokines, released from M2 macrophages, inhibit osteoclastogenesis and promote Th2 profile and GCs formation [15,92,157,158]. In particular, M2a fusion under the presence of E-Cadherin, DC-STAMP and MR produce FBGCs. M2bλ fusion form GCs during tuberculosis thanks to MR and other potential fusogens like TLR and MyD88. M2d fusion under not well-defined fusogenic molecules favours the formation of TGCs [139]. Conversely, the inflammatory mediators secreted by M1 macrophages mainly display an osteoclastogenic potential. They can substitute RANKL activity by promoting Th1/Th17 profile and OCs formation, thereby deregulating bone remodelling [7,15,159,160]. In particular, these cytokines support pre-OCs recruitment to the bone microenvironment and their differentiation into OCs [161]. Moreover, they contribute to generating LGCs, often detected in chronic inflammatory disorders like RA [139]. Among inflammatory mediators, TNF-α is a potent inflammatory inducer of bone resorption. TNF-α, in synergy with RANKL, can activate excessive osteoclastogenesis and bone resorption. It enhances RANK expression on OCs precursors and M-CSF and RANKL in OBs and synovial fibroblasts [159]. In RA, TNF-α also acts on synovial cells by stimulating IL-34, NF-κB and JNK signalling [162]. IL-1β is another osteoclastogenic cytokine, which directly stimulates OCs differentiation under sufficient levels of RANKL. IL-1β can still foster the differentiation of pre-OCs by enhancing TNF-α-induced osteoclastogenesis and RANKL expression [15]. Interestingly, IL-1β can promote distinct processes of multinucleation of bone marrow OCs precursors. Following IL-1β stimulation, CD-31^+^ Ly-6C^+^ myeloid blasts show the fast production of OCs (>20 nuclei) with a top-level of bone resorption and shortened lifespan. CD-31-, Ly-6 Chi monocytes display a lower number of OCs formation when compared to CD-31+ Ly-6C+ myeloid blasts but higher life span [163]. IL-1 β supports most large OCs by increasing cyclin D in early blast cultures [163]. Interleukin-1 receptor-associated kinase 4 (IRAK4) is a protein kinase, which transduces signals from inflammatory cytokines and toll-like receptors by stimulating natural killer cells, antigen-presenting cells and T-cells [143]. In inflammatory conditions like IL-1β stimulation, IRAK-deficient cells display reduced osteoclastogenesis and enhanced GCs formation [143]. In this light, IRAK-4 might be a therapeutic target to modulate M1/M2 polarisation by antagonising inflammatory osteolysis. Using a system genetic approach, Behmoaras J et al. identified a trans-regulated multinucleation network in macrophages with a critical role of KCNN4 in bone homeostasis and inflammatory disorders [6]. KCNN4 regulates Ca^2+^ signalling during macrophage multinucleation. KCNN4^-/-^ mice with collagen antibody-induced arthritis (CAIA) model showed reduced joint inflammation, tissue damage and serum bone resorption markers without affecting OBs activity; thus, opening fascinating insights to prevent inflammation-related bone loss [6]. Over the last years, there has been an increasing awareness that mononuclear phagocytes display receptors for extracellular nucleotides, which modulate inflammatory responses. P2RX purinergic receptors belong to this subfamily and bind to the extracellular adenosine 5’-triphosphate (ATP). P2RX7 promotes several physiologic and pathologic conditions, including the multinucleation of monocyte-derived human macrophages [164]. Macrophages with high levels of P2RX7 are more prone to form GCs rather than OCs [165]. P2RX7 contributes to activating the inflammasome in both OCs and GCs. P2RX7 favours the release of IL-1β and IL-18 and the synthesis of ROS with essential implications in caspase activation and apoptosis induction [166]. Similarly, P2RX5 promotes OCs-mediated inflammatory bone loss and hyper-multinucleation of OCs [77]. During inflammatory conditions, P2RX5^-/-^ mice display bone loss because of the P2RX5-mediated inflammasome activation and IL-1 β production, important for OC maturation [77]. Interestingly, several miRNA, including miR-9, miR-127, miR-155 and miR-125b would seem implicated in promoting the activation of M1 macrophages and pro-inflammatory responses by targeting several adaptor proteins and transcription factors [167] (Table 3).

### 3.3. Pathological Features of RA

RA is one of the most widespread chronic inflammatory and autoimmune illnesses with a tremendous impact on patients quality life, leading to joint swelling, pain, and destruction [168]. Risk factors include genetic determinants (e.g., MHC class II genes, HLA-DR1, HLA-DR2, HLA-DRB1), smoking, obesity, infections [169,170] and female gender (with a two-fold increase risk than male) [171]. Being an autoimmune disorder, RA patients display an excessive immune response of T-cells, leading to the formation of autoantibodies. Th17 cells are a pro-osteoclastogenic T-cell subset, markedly activated than Treg with a critical functional role in inflammation. In particular, Th17 cells activate several immune cells and foster OCs activity by inducing RANKL in synovial fibroblasts [168]. This excessive immune response leads to high expression of rheumatoid factor (RF) and anti-citrullinated protein antibodies (ACPAs), considered as a powerful predictor of bone erosion [172]. ACPAs are the most specific biomarker found in RA serum, capable of mediating bone loss [5,173,174].

The inflamed synovial membrane, characterised by the presence of T cells, synovial fibroblasts, activated macrophages and excessive angiogenesis processes, is a typical feature in RA patients, which culminates in joint destruction [175]. Activated T cells and arthritic synovial fibroblasts provide alternative sources of RANKL throughout the synovial lining layer near sites of bone erosion [176]. Increased RANKL mRNA and protein expression in the synovium and lowered expression of OPG occur in RA patients. The altered RANKL/OPG ratio at the pannus-bone interface contribute to supporting focal lesions in RA; thus representing a potential target for therapeutic intervention [161]. In RA synovium, macrophages play a pivotal role in feeding the inflammatory network [152]. The percentage of pro-inflammatory M1 is higher than M2 subset due to the excessive activation, synovial proliferation and enhanced anti-apoptotic activities [152]. The newly inflammatory milieu plays a pivotal role in promoting resorption activities and reducing bone formation. In this light, TFN-α and IL-1β modulate pre-OCs recruitment to the inflamed sites and their differentiation into OCs and inhibit OBs differentiation [161]. Inhibitors like DKK1, DKK3, sFRP1, sFRP2, and sFRP4 impair bone formation at erosion sites leading to reduced canonical wingless (Wnt) pathway, crucial for bone deposition, development and remodelling [161]. Beyond activated macrophages, RA synovium displays distinct patterns of GCs subtypes, including LGCs and TRAP^+^/CTSK^−^ FBGCs. Such cells can play several roles, including antigen presentation and promotion of OCs differentiation [15]. Local accumulation of TRAP^+^ and CTSK^+^ OCs in the articular joint triggers erosion processes on both bone and articular cartilage [15] (Figure 5). Besides activating inflammatory and catabolic pathways, the cytokine “storm” impairs oxidant/antioxidant balance in joint tissues, which is another hallmark of the disease responsible for tissue destruction [177,178]. RA patients display a five-fold increase in mitochondrial ROS production in whole blood and monocytes when compared to healthy subject [179]. M1 macrophages produce high levels of ATP and ROS because of excessive glucose uptake [152]. Another factor recently implicated in the pathogenesis of RA is the deregulation of the autophagic pathway, a pro-survival mechanism responsible for cell responses following injury. The mammalian target of rapamycin (mTOR) complex 1 (mTORC1) activates the autophagy mechanism. In RA, its impairment promotes (i) osteoclastogenesis; (ii) the survival of inflammatory cells, and (iii) the generation of citrullinated peptides [180]. Autophagy also participates in the reduction of ROS through the elimination of damaged mitochondria to prevent apoptosis [181].

### 3.4. Therapeutic Strategies in RA

Identifying potential new therapeutic candidates for RA is a challenge among clinicians because of the multi-coloured clinical scenario of patients presenting structural and functional joint alterations and many systemic effects [177]. Despite their promising clinical outcomes, the use of anti-resorptive drugs targeting OCs is inadequate [182]. Currently, disease-modifying anti-rheumatic drugs (DMARDs) are among the first-line strategies for RA treatment. This class of drugs target inflammatory reactions [177]. An alternative classification system distinguishes conventional synthetic chemical compounds (csDMARDs) and targeted synthetic DMARD (tsDMARDs) [183]. Despite the effects of DMARDs on the phlogistic environment, they display several contraindications and side effects, as they interfere with the immune system by enhancing the susceptibility to infections [175]. Accordingly, research efforts generated a new class of drug, biological DMARDs (bDMARDs) treatments, which envisage the use of biological agents [184,185]. (Figure 6). Methotrexate (MTX), firstly used for RA treatment, inhibits the proliferation of inflammatory synovial cells and reduce macrophage and lymphocyte recruitment functions [175]. JAK inhibitors are a class of tsDMARDs, which block the JAK/STAT signalling pathway involved in the signalling transduction of several cytokines [186]. This pathway, composed of JAK1, JAK2, JAK3, and the tyrosine kinase 2 (TyK2), mediates intracellular signals through the transcription factor, STAT [187]. Tofacitinib and baricitinib are two JAK inhibitors approved by the Food and Drug Administration/European Medicines Agency (FDA/EMA) for RA treatment, which impair T lymphocyte RANKL production but not OCs differentiation and function [188] and promote bone formation and repair [189]. Autophagy regulates the innate and adaptative immune system and plays a crucial role in osteoclastogenesis [180]. In an arthritis model, the inhibition of autophagy contributed to reducing bone erosion and OCs number; suggesting its potential role in bone degradation [190]. In this light, drugs lowering autophagy might be another alternative to prevent bone resorption. bDMARDs can be divided into different subfamilies depending on their biological targets. They can include monoclonal antibodies (mAbs) and modified proteins targeting either cytokines or cell-surface molecules [191]. Inhibiting inflammatory cytokine with an osteoclastogenic profile can be a valid approach for blocking OCs-mediated bone resorption; thus, preventing focal bone loss. TNF inhibitors were among the first bDMARDs. Over the last years, scientists developed five main TNF-α inhibitors [192]. Tocilizumab is a humanised monoclonal antibody specific to IL-6R used as monotherapy or with MTX [161]. bDMARDs can also target cell surface molecules. In this light, modulating RANKL/OPG ratio at the pannus-bone interface can prevent bone erosions. Denosumab is a human monoclonal IgG2 antibody against RANKL, which suppress osteoclastogenesis by lowering bone resorption but with no effects on inflammation and cartilage erosion [176]. Targeting cell sources producing inflammatory mediators rather than their final products could be an alternative option due to the growing knowledge of molecules and signalling pathways associated with M1 and M2 [193,194,195,196].

#### Future Therapeutic Perspectives in RA Treatment

Despite remarkable progress in treatment modalities, there is still an essential demand for establishing new therapeutic targets. Being a multi-coloured disorder, a broad spectrum of alternative options, focused on inflammation, M-FM and osteoclastogenesis could be considered to prevent RA evolution.

First, targeting the inflammatory circuit, mainly supplied by activated macrophages, could envisage several alternatives: (i) neutralising major inflammatory cytokines via the use of inhibitors; and (ii) modulating the activation status of macrophages. Accordingly, clinical trials on TNF blockers [197], IL-1 [198], and IL-6 receptor blockage [199,200] gave evidence of the impact of inflammation on osteoclastogenesis by retarding or inhibiting bone erosion in RA patients. On the other hand, the use of selective agents switching the M1 towards M2 phenotype could improve RA mitigating the severe, continuous and debilitating pains symptoms. In this light, natural compounds, like triterpenoids, stilbenes, flavonoids, several miRNA and gene editing approaches could contribute to modulating macrophage phenotype [167,185,201,202]. IRAK-4 might be another therapeutic target to modulate M1/M2 polarisation by antagonising inflammatory osteolysis. Despite their promising results, further preclinical and clinical studies are necessary to define better what natural products and miRNAs are more relevant and elucidate what their signalling pathways are.

Second, targeting macrophage fusion and multinucleation could be a strategic alternative to avoid an excessive formation of OCs and GCs and enhanced inflammatory reactions, implicated in bone erosion and joint destruction. The technological advances could help to unravel conventional fusion mediators during the early stages by exploiting their pre-fusion transcription profile to develop new therapeutic strategies [7]. Inhibitors of multinucleation, already listed for OP, could be shared for treating especially bone loss in RA. Moreover, targeting KNN4, implicated in macrophage multinucleation [6], could be a potential alternative to ensure prevention of both inflammation and bone loss simultaneously to prevent inflammation-related bone loss. Notably, regulating P2RX7 expression on GCs and OCs could open essential perspectives for blocking the inflammatory circuit [203,204]. To this end, some scientists demonstrated through in vitro studies that its inhibition can prevent M-CSF/RANKL stimulated fusion of human monocytes, thereby inhibiting OCs multinucleation, however, its mechanism of action is still unknown [205].

Identifying the specific fusogenic molecules generating OCs and GCs side by side and clarifying the subtypes of GCs residing in the joint tissues thanks to the technological advances would be instrumental for designing more targeted strategies [7,167,206] (Table 4).

Third, getting a better understanding of the influence of anti-rheumatic drugs on microbiota and their subsequent effects on the immune system might be instrumental for selecting drugs with limited side effects on immune cells. Combining standard treatments with probiotics or natural compounds might be an alternative for inhibiting typical catabolic and inflammatory processes by preserving the composition and function of the microbiota in RA patients, which often display dysbiosis [207,208].

The knowledge from studies addressing these aspects will be instrumental for improving current therapeutic options and redesigning more targeted approaches in the future depending on the stage of disease severity.

## 4. Conclusions

OP and RA are worldwide concerns, sharing some clinical and biological features of bone loss because of the alteration of the bone remodelling process, with an enormous burden for the health care system. Therefore, identifying and developing effective and therapeutic options is paramount to diseases treatments. Controlling aberrant inflammatory signals while preserving bone homeostasis is a significant challenge for both OP and RA. Interestingly, the macrophage multinucleation could be a biological target being a phenomenon perturbed in bone and inflammatory disorders, thereby opening valuable therapeutic insights in both RA and OP. In this light, targeting specific fusogenic determinants, like DC-STAMP, Siglec-15, KNN4, P2XR7, implicated in either OCs or GCs formation could provide alternative strategies to inhibit the inflammatory and degenerative processes in RA and OP, which culminate in joint destruction. Targeting a particular cell macrophage subset could be a valid strategy to regulate their balance in these disorders, because of the influence of M1 and M2 macrophages on OCs and GCs formation. This review provides an up-to-date overview of M-FM in generating OCs and GCs in both disorders and the ongoing researches on the role of inflammation in driving the heterogeneous and dynamic macrophage phenotype and the formation of mature multinucleated. However, further in-depth studies focused on the mechanism and timing of M-FM, the fusion machinery in OCs and GCs biology are necessary for better elucidating their role in OP and RA.

## Figures and Tables

**Figure 1 ijms-21-06001-f001:**
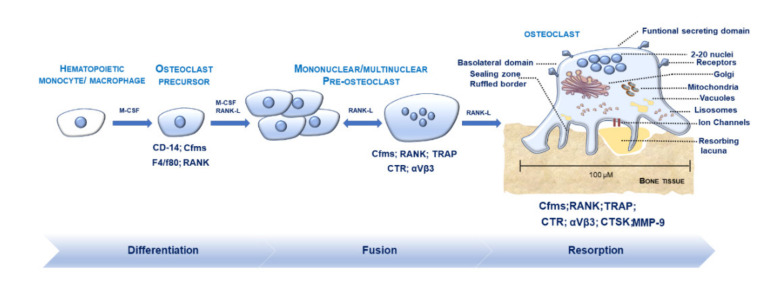
Osteoclasts (OCs) formation and differentiation. Hematopoietic monocytes/macrophages mature into OCs precursors (pre-OCs) (positive for CD-14, Cfms, F4/F80, RANK) after macrophage colony-stimulating factor (M-CSF) stimulation. The addition of M-CSF and receptor activator for nuclear factor κ B ligand (RANK-L) drives differentiation of pre-OCs and fusion towards mature multinucleated OCs (positive for Cfms, RANK, TRAP, CTR, CTSK, MMP-9). OCs display a polarised shape and own up to 2–20 nuclei. OCs surface membrane exhibits channels (responsible for the release of ions and matrix-degrading enzymes favouring bone resorption) and four distinct domains: the sealing zone (SZ), the ruffled border (RB), the basolateral domain (BD), and the functional secretory domain (FSD). Golgi apparatus, mitochondria, lysosomes, vacuoles are within the cytoplasm to support OCs function.

**Figure 2 ijms-21-06001-f002:**
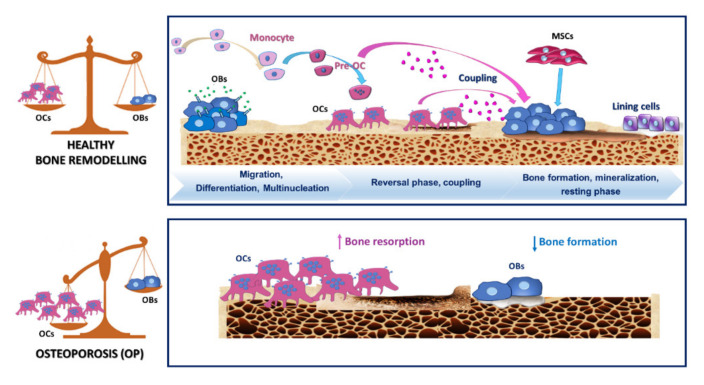
Graphical depiction of processes underpinning normal bone remodelling and osteoporosis (OP). The upper panel illustrates the classic phases occurring during normal bone remodelling, including migration of monocyte/macrophage precursors and differentiation towards osteoclasts precursors (pre-OCs) and mature multinucleated osteoclasts (OCs). Bone remodelling also requires a cross-talk between osteoblasts (OBs) and OCs to ensure a proper equilibrium between bone resorption and production. The lower panel describes the altered balance between bone resorption and formation in favour of bone resorption by OCs.

**Figure 3 ijms-21-06001-f003:**
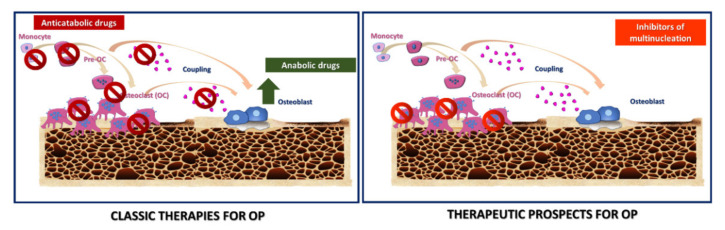
Graphical representation of classic and new therapeutic prospects for OP. The left panel illustrates standard remedies for OP, divided into two classes: anti-catabolic and anabolic drugs. The first class inhibits bone resorption by targeting either the differentiation, resorptive function, cytokines production by osteoclasts (OCs). The second class builds up bone architecture, stimulating osteoblasts (OBs) and their precursors. The right panel outlines new therapeutic prospects for OP. It refers to inhibitors of multinucleation, which would preserve the OCs precursors to support the OCs-OBs coupling.

**Figure 4 ijms-21-06001-f004:**
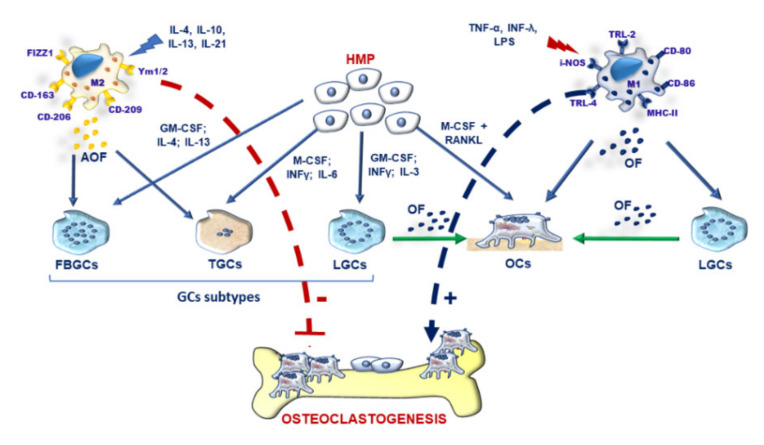
Graphical representation of factors modulating osteoclasts (OCs) and giant cells (GCs) formation. M2 macrophages, induced by IL-4, IL-10, IL-13 and IL-21, can release anti-osteoclastogenic factors (AOF), which promote foreign-body giant cells (FBGCs) and Touton giant cells (TGCs). Hematopoietic monocyte/macrophages precursors (HMP) under (i) GM-CSF, IL-4 and IL-13 can originate FBGCs; (ii) M-CSF, INF-γ, and IL-6 stimuli can induce Touton cells (TGCs); and (iii) GM-CSF, INF-γ and IL-3 can produce Langhans giant cells (LGCs). FBGCs, TGCs and LGCs represent the main three subtypes of GCs, displaying distinct histological and functional features. HMP can originate mature OCs after M-CSF and RANKL stimulation. M1 macrophage, induced by TNF-α, LPS and INF-λ, release osteoclastogenic factors (OF), which promote OCs and LGCs. LGCs can release OF, which in turn support the OCs differentiation. M2 macrophages inhibit osteoclastogenesis by releasing AOF (red arrow). M1 macrophages promote osteoclastogenesis via the release of (blue arrow).

**Figure 5 ijms-21-06001-f005:**
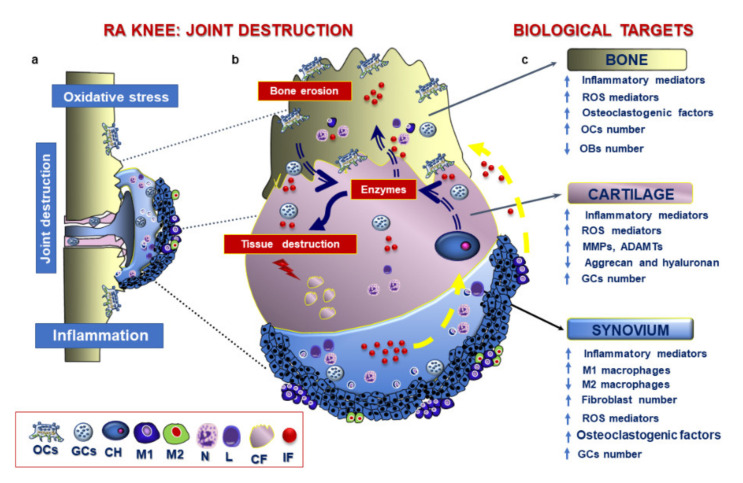
Graphical representation of processes in RA knee and potential biological targets. (**a**) RA displays inflammatory, catabolic and oxidative processes in the synovial membrane, subchondral bone and articular cartilage. (**b**) A high-magnification view of processes implicated in joint destruction. Synovial membrane displays hyperplasia of the lining layer and marked activation of neutrophils, lymphocytes and M1 macrophages, which release inflammatory factors (IF). The phlogistic environment in the synovial membrane (yellow arrow) fosters the release of several matrix-degrading enzymes by chondrocytes (CH) and the release of osteoclastogenic cytokines by giant cells (GCs). These factors trigger the destruction of both bone and cartilage fragments (CF). The inflammatory milieu still fed by GCs amplify this vicious circuit (**c**) List of main biological targets in the bone, cartilage and synovium. OCs: osteoclasts; GCs: giant cells; CH: chondrocytes; M1: M1 macrophages; M2: M2 macrophages; N: neutrophils; L: lymphocytes; CF: cartilage fragments; IF: inflammatory factors.

**Figure 6 ijms-21-06001-f006:**
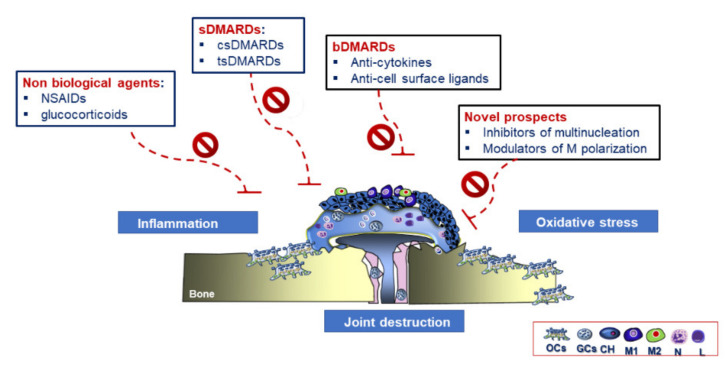
Graphical representation of standard treatments and new prospects for RA. Non-biological agents, synthetic disease-modifying anti-rheumatic drugs (sDMARDs), and biological disease-modifying anti-rheumatic drugs (bDMARDs) are among the main standard treatments for RA patients. Using inhibitors of multinucleation and modulators of macrophages (M) polarisation could offer alternative novel strategies in RA.

**Table 1 ijms-21-06001-t001:** Biological targets involved in monocyte-macrophages fusion and multinucleation (M-FM).

Biological Target	Role in M-FM	Refs
CD44/Matrix metallopeptidase 9 (**CD44/MMP-9**)	▪ Enhances the motility signals for stimulating cells to migrate and fuse.	[45]
Monocyte chemoattractant protein-1/C-C chemokine receptor type 2 (**MCP-1/CCR2**)	▪ Crucial for the formation of mature multinucleated OCs.	[49,50]
**CD9**	▪ Permissive fusogen.	[51]
Dendritic cell-specific transmembrane protein (**DC-STAMP**)	▪ OCs-specific fusogen.	[37,53]
OC-stimulatory transmembrane protein (**OC-STAMP**)	▪ OCs-specific fusogen.	[44]
**Syncitin-1**	▪ Drives the fusion of the plasma membranes lipid bilayers. ▪ Drives the fusion between multinucleated cells rather than mononuclear pre-OCs.	[58]
Sialic acid-binding immunoglobulin-type lectin 15 (**Siglec-15**)	▪ Key to the formation of the actin ring. ▪ Key to the formation of multinucleated OCs.	[59,60,61,62,63]
Ras-related protein Rab-27a (**Rab27a**)	▪ Mediates lysosomes trafficking and membrane fusion. ▪ Regulates the transport of LRO to modulate multinucleation and cell size in OCs.	[64]
Osteoclastogenesis-associated transmembrane protein-1 (**Ostm1**)	▪ Inhibits M-FM by targeting NFATc1.	[65]
**miR7b**	▪ Targets and inhibits DC-STAMP.	[37]
**miR30a**	▪ Targets and inhibits DC-STAMP.	[35]
**miR-26a**	▪ Targets CTGF/CCN2 and inhibits DC-STAMP.	[66]
**CD47**	▪ Key to the fusion of two mono-nucleated partners or mono- and multinucleated partners. ▪ Promotes the formation of large OCs and reduces the formation of smaller OCs.	[58]
Macrophage fusion receptor (**MFR**)	▪ Plays a role in macrophage-macrophage adhesion/fusion leading to multinucleation.	[7,10]
**E-cadherin**	▪ Drives the formation of dynamic membrane protrusions necessary for migration and fusion. ▪ Promotes the formation of multinucleated OCs.	[7]
**CD-26**	▪ Key to the formation of multinucleated OCs.	[68]
**CD-47**	▪ Key to the fusion of two mono-nucleated partners or mono- and multinucleated partners. ▪ Favour the formation of large OCs and to reduce the formation of smaller OCs.	[69]
Src non-receptor tyrosine kinase (**c-Src**)	▪ Maintains the dynamic organization of the ZLS. ▪ Key to the formation of multinucleated OCs.	[22,73]
Human protein ‘SH3 and PX domains 2A’ (**Tsk5)**	▪ Promotes the formation of podosomes and fusion-competent protrusions.	[24]
C-C chemokine receptor type 1 (**CCR-1)**	▪ Key to the cell fusion.	[22]
Rapamycin-insensitive companion of TOR (**RICTOR**)	▪ Regulates OCs fusion by up-regulating DC-STAMP.	[23]
Tenascin x (**TNX**)	▪ Suppresses OCs multinucleation.	[78]
Dynamin	▪ Key to the formation of multinucleated OCs.	[79]
Two-pore channel 2 (**TPC2**)	▪ Downstream effector of RANKL involved in differentiation, multinucleation.	[80]
Fibronectin leucine-rich transmembrane protein 2 (**Flrt2**)	▪ Key to the formation of multinucleated OCs.	[81]
Calcium release-activated channels (**CRAC-C channels**)	▪ Key to the formation of multinucleated OCs.	[82]
**Transcription factor Spi-C** (**SPIC**)	▪ Governs both early and late stages of OCs differentiation among which multinucleation and bone-resorbing functions.	[83]
Crk-associated substrate (**Cas**)	▪ Key to actin cytoskeletal reorganization, actin ring formation and multinucleation of OCs	[85]
**Luman**	▪ Regulates the expression, localization and stability of DC-STAMP.	[70]
Vacuolar ATPase (**ATP6v0d2**)	▪ Key to the formation of multinucleated OCs.	[71]
**DAP-12**	▪ Key for acquiring fusion competence. ▪ Key to the formation of multinucleated OCs.	[72]
**OSCAR-FcRy**	▪ Key for acquiring fusion competence.	[73]
**Transglutaminases**	▪ Regulates migration and fusion of pre-OC.	[74]
**P2 × 7**	▪ Key to the formation of multinucleated OCs in vitro.	[75]
**P2 × 5**	▪ Key to the formation of multinucleated OCs in vitro.	[76,77]
**miR124**	▪ Targets and inhibits Rab27a.	[67]

**Table 2 ijms-21-06001-t002:** Perspectives for osteoporosis (OP) therapy: biological targets for selective inhibition of monocyte-macrophage fusion and multinucleation (M-FM) in osteoclasts (OCs).

Biological Targets	Therapeutic Molecules/Compounds	Effects on M-FM and OP	Refs
**CD9**	**Anti-CD9 antibody**	▪ Inhibits the multinucleation in vitro. ▪ Suppresses bone resorption in a mouse model of periodontitis.	[51]
**DC-STAMP**	**Lentiviral vector pre-miR-7b**	▪ Mediates the overexpression of miR7b in OCs/ ▪ Inhibits the protein expression of DC-STAMP. ▪ Inhibits the multinucleation in vitro.	[37]
	**MiR-30a mimic**	▪ Mediates the overexpression of miR7b in OCs. ▪ Inhibits the protein expression of DC-STAMP. ▪ Inhibits the multinucleation in vitro.	[135]
	**anti-DC-STAMP-monoclonal antibody**	▪ Inhibits the multinucleation in vitro. ▪ Reduces the number of multinucleated TRAP+ cells and alveolar bone loss in a mouse model of periodontitis.	[136]
**OC-STAMP**	**anti-OC STAMP-monoclonal antibody**	▪ Inhibits the multinucleation in vitro. ▪ Reduces the number of multinucleated TRAP^+^ cells and alveolar bone loss in a mouse model of periodontitis.	[44]
**CTGF/CCN2**	**miR-26a mimics**	▪ Mediates the overexpression of miR26a in OCs. ▪ Inhibits the protein expression of CTGF/CCN2 and DC-STAMP. ▪ Inhibits the multinucleation in vitro.	[66]
**Siglec-15**	**anti-Siglec-15 antibody**	▪ Inhibits the multinucleation in vitro. ▪ Induced increased bone mineral density in young mice. ▪ Attenuates osteoporosis by impairing OCs function but not skeletal growth in young mice. ▪ Protects against glucocorticoid-induced OP of growing skeleton in juvenile rats.	[60,63,137,138]
**Rab27a**	**MiR-124 mimics**	▪ Mediates the overexpression of miR124 in OCs. ▪ Inhibits the protein expression of Rab27a. ▪ Inhibits the multinucleation in vitro.	[67]

**Table 3 ijms-21-06001-t003:** Biological targets involved in monocyte-macrophages fusion and multinucleation (M-FM) during the inflammatory bone loss.

Biological Target	Effects on M-FM and Inflammatory Bone Loss	Refs
Macrophage fusion receptor **(MFR)**	▪ Promotes macrophage fusion. ▪ Promotes the differentiation of OCs and GCs.	[144,145]
Potassium calcium-activated channel subfamily N member 4 **(KCNN4)**	▪ Favours the macrophage fusion and multinucleation in bone homeostasis and inflammatory disorders. ▪ Promotes the differentiation of OCs and GCs.	[6]
Tetraspanins **(CD-9, CD81, CD63, CD53)**	▪ Promote monocyte/macrophage fusion. ▪ Release several osteoclastogenic cytokines. ▪ Promote the fusion of pre-OC towards mature OCs.	[150]
Pro-inflammatory macrophages **(M1)**	▪ Promotes osteoclastogenesis through the release of osteoclastogenic cytokines. ▪ Promotes the induction of Th1/Th7 profile. ▪ Promotes the formation of Langherans giant cells (LGCs). ▪ Favours the production of reactive oxygen species (ROS).	[15,98,152,153]
Interleukin-1 beta **(IL-1β)**	▪ Promotes the multinucleation of OCs and GCs ▪ Favours the differentiation and maturation of large OCs. ▪ Promotes the pre-OCs differentiation via TNF-α-induced osteoclastogenesis. ▪ Promotes the multinucleation of bone marrow precursors.	[7,163]
Tumour necrosis factor-alpha **(TNF-α)**	▪ Promotes osteoclastogenesis. ▪ Promotes in synergy with RANKL excessive osteoclastogenesis and bone resorption. ▪ Fosters RANKL and M-CSF in synovial fibroblasts and osteoblasts.	[159]
Interleukin 6 (**IL-6**)	▪ Stimulates OCs maturation. ▪ Promotes vascular endothelial growth factor (VEGF)-stimulated pannus proliferation. ▪ Promotes synovitis and joint destruction. ▪ Promotes B-cell maturation and TH-17 differentiation.	[152,158]
Signal transducer and activator of transcription-6/-1 axis **(STAT-6/STAT-1 axis)**	▪ Regulates OC-STAMP and DC-STAMP. ▪ Regulates fusogenic mechanisms in FBGCs.	[7,148]
Tumour necrosis factor receptor-associated factor 6 **(TRAF6)**	▪ Regulates inflammatory responses. ▪ Regulates the differentiation of various immune cells. ▪ Promotes the macrophage polarization into M2 subset.	[156,160]
Interleukin-1 receptor-associated kinase 4 **(IRAK4)**	▪ IRAK-deficient cells display reduced osteoclastogenesis and enhanced GCs formation.	[143]
Purinergic receptor P2X7 (**P2RX7)**	▪ Promotes the multinucleation of monocyte-derived human macrophages. ▪ Activates the inflammasome in both OCs and GCs. ▪ Promotes the release of inflammatory and ROS molecules.	[164,165,166]
Purinergic receptor P2X5 (**P2RX5)**	▪ Promotes OCs-mediated inflammatory bone loss. ▪ Promotes hyper-multinucleation of OCs. ▪ Promotes inflammasome activation and IL-1 β production.	[76]
**Mammalian target of rapamycin (mTOR) complex 1 (mTORC1)**	▪ Activates the autophagy. ▪ Is implicated in regulating bone resorption and homeostasis in pathologic conditions.	[23,101]
**miR9**	▪ Promotes M1 polarization through targeting the peroxisome proliferator-activated receptor δ (PPARδ). ▪ Overexpression of mir9 prevents the BCL-6- mediated anti-inflammatory effects.	[167]
**miR127**	▪ Is prominently induced upon toll-like receptor (TLR) engagement. ▪ Enhances the activation of JNK Kinase and the development of M1 macrophages. ▪ Promotes the production of pro-inflammatory cytokines.	[167]
**miR125b**	▪ Promotes the formation of M1 macrophages. ▪ Promotes pro-inflammatory responses.	[167]
**miR155**	▪ Targets IL-13 receptor α1. ▪ Inhibits STAT-6 activation. ▪ Promotes M1 polarisation.	[167]
**Dysbiosis**	▪ Impairs immune response. ▪ Promotes various pro-inflammatory signalling pathways.	[97]

**Table 4 ijms-21-06001-t004:** Perspectives for rheumatoid arthritis (RA) therapy: biological targets for inhibiting inflammation and monocyte/macrophage fusion and multinucleation (M-FM) in osteoclasts (OCs) and giant cells (GCs).

Biological Targets	Therapeutic Molecules/Compounds	Effects on M-FM and Inflammation in RA	Refs
**Tetraspanins**	**Anti-tetraspanins antibodies**	▪ Inhibit fusion rate, and the size of CGs obtained from intermediate monocyte subset.	[150]
Tumour necrosis factor-alpha (**TNF-α**)	**TNF blockers**	▪ Prevent the loss of bone mineral density in RA patients. ▪ Reduce serum levels of carboxy-terminal telopeptide of type 1 collagen and RANKL in RA patients. ▪ Prevent spine and hipbone loss.	[192,197]
Interleukin-1 (**IL-1**)	**IL-1 inhibitors**	▪ Improve both glycaemic and inflammatory parameters in patients with RA and type II diabetes.	[198]
Interleukin-6 receptor (**IL-6R**)	**IL-6R blockers**	▪ Are effective treatments in phase III clinical trials in RA patients.	[199,200]
**JAK/STAT cascade**	**JAK inhibitors** **(JAK1, JAK2, JAK3)**	▪ Inhibit transduction signal from type I and II cytokine receptors. ▪ Reduce inflammatory-mediated effects. ▪ Tofacitinib and baricitinib are two JAK inhibitors approved by FDA/EMA for RA patients.	[187,188,189]
**P2X7 signalling**	**Anti-P2X7 antibody**	▪ Prevents M-CSF/RANKL stimulated fusion of human monocytes.	[205]
**Mir127**	**Antagonist of mir127**	▪ Modulates macrophage polarization in favour of M2 macrophage subset. ▪ Reduces the expression of osteoclastogenic cytokines.	[167,202]
**Macrophage polarization**	**Nutraceuticals** **(tripertenoids, stilbenes, flavonoids**)	▪ Promote macrophage polarization toward wound-healing M2 activation status. ▪ Prevent inflammatory osteolysis.	[185]
**Dysbiosis**	Disease-modifying anti-rheumatic drugs **(DMARDs)**	▪ Target cytokines, nonspecific immune suppression or T-cell and B-cell activation. ▪ Interfere in various pro-inflammatory signalling pathways. ▪ Promote partial restoration of eubiotic gut microbiota.	[207,208]

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
