# Peer review of "Learning from Monocyte-Macrophage Fusion and Multinucleation: Potential Therapeutic Targets for Osteoporosis and Rheumatoid Arthritis"

_ijms, 2020, doi:10.3390/ijms21176001_

Round 1

Reviewer 1 Report

This review attempted to explain bone resorption mechanisms in terms of monocyte-macrophage fusion and multinucleation mechanisms in osteoporosis and rheumatoid arthritis patients. The manuscript was well-organized and included proper previous literatures and graphics.

I have one suggestion to improve the readability of this study. The main goal of this study was to suggest the new therapeutic targets for osteoporosis and rheumatoid arthritis based on osteoclast acitivity induced by multinucleation methanisms. Please provide tables which summarized the main target, gene, and pathway of those new targets to improve readability for authors.

Author Response

We thank reviewer 1 for providing useful tips and comments to improve the quality of the paper and for better comprehension for the readers.

Reviewer 1

This review attempted to explain bone resorption mechanisms in terms of monocyte-macrophage fusion and multinucleation mechanisms in osteoporosis and rheumatoid arthritis patients. The manuscript was well-organized and included proper previous literature and graphics.

-) We thank the Reviewer for his/her comments. We added further bibliographic references and supplementing information, as suggested by reviewer 2. In particular, we added details in the manuscript about: i) the role of ephrins in osteoclastogenesis (Page 3 from Line 144 to 151); ii) the role of immune systems in the OP (Page 7 from Line 277 to 279 and on Page 8 from Line 297 to 310); and iii) indications of romosozumab among anabolic agents (Page 9; Lines, 382-383). Moreover, we corrected some spelling errors within the document.

I have one suggestion to improve the readability of this study. The main goal of this study was to suggest the new therapeutic targets for osteoporosis and rheumatoid arthritis based on osteoclast acitivity induced by multinucleation mechanisms. Please provide tables which summarized the main target, gene, and pathway of those new targets to improve readability for authors.

-) We thank the Reviewer for his helpful suggestion to improve the quality of this manuscript. We agree with him that tables summarising the main therapeutic targets and their mode of action can be instrumental in enhancing readability for authors. We added four tables into the document. In Table 1, we described biological targets involved in monocyte-macrophages fusion and multinucleation (M-FM) (Pages 5, 6 and 7). In Table 2, we described perspectives for osteoporosis (OP) therapy: biological targets for selective inhibition of monocyte-macrophage fusion and multinucleation (M-FM) in osteoclasts (OCs) (See Pages 10 and 11). In Table 3, we described biological targets involved in monocyte-macrophages fusion and multinucleation (M-FM) during the inflammatory bone loss. (See Pages 14, 15 and 16). In Table 4, we described Perspectives for rheumatoid arthritis (RA) therapy: biological targets for inhibiting inflammation and monocyte/macrophage fusion and multinucleation (M-FM) in osteoclasts (OCs) and giant cells (GCs) (See pages 19 and 20).

Reviewer 2 Report

The review is interesting and well written, few items swhould be addressed:

  1. the role of Ephrins in osteoclastogenesis should be described;
  2. the role of immune systems in the pathogenesis of osteoporosis should be detailed (Pacifici R papers), togheter with the involved molecules i.e. J Pathol. 2020 Apr;250(4):440-451.
  3. About monoclonal antibodies towards osteoporosis the following reference should be added: Expert Opin Biol Ther. 2018 Feb;18(2):149-157. Furthermore, as anabolic agent, the use of romosozumab should be reported.

Author Response

We thank reviewer 2 for providing helpful suggestions and comments to improve the quality of the manuscript. We added four tables into the document, as recommended by reviewer 1. In Table 1, we described biological targets involved in monocyte-macrophages fusion and multinucleation (M-FM) (Pages 5, 6 and 7). In Table 2, we described perspectives for osteoporosis (OP) therapy: biological targets for selective inhibition of monocyte-macrophage fusion and multinucleation (M-FM) in osteoclasts (OCs) (See Pages 10 and 11). In Table 3, we described biological targets involved in monocyte-macrophages fusion and multinucleation (M-FM) during the inflammatory bone loss. (See Pages 14, 15 and 16). In Table 4, we described Perspectives for rheumatoid arthritis (RA) therapy: biological targets for inhibiting inflammation and monocyte/macrophage fusion and multinucleation (M-FM) in osteoclasts (OCs) and giant cells (GCs) (See pages 19 and 20). Moreover, we corrected some spelling errors within the document.

Reviewer 2

  1. The role of Ephrins in osteoclastogenesis should be described.

-) We thank the reviewer for his/her kind suggestion to improve the quality of the manuscript. We added further details on the role of ephrins in osteoclastogenesis. We inserted the text reported below.

Key signalling modulating osteoclastogenesis derives from cell-cell contacts mediated by proteins such as ephrins. Interestingly, cell-cell communication between OCs and OBs through ephrinA2 (expressed in OCs downstream to RANKL signalling)-EphA2 (expressed in OBs) activates the initiation phase of bone remodelling by enhancing OCs differentiation and suppressing OBs differentiation [39]. Conversely, cell-cell contact between OCs and OBs through ephrinB2 (expressed in mature OCs)-EphB4 (expressed in OBs precursors) enhances osteogenic differentiation and suppresses OCs function [39].

You can find all these details on Page 3 from Line 144 to 151.

  1. The role of immune systems in the pathogenesis of osteoporosis should be detailed (Pacifici R papers), together with the involved molecules, i.e. J Pathol. 2020 Apr;250(4):440-451.

-)  We thank the reviewer for his/her kind suggestion to improve the quality of the manuscript. We added details on the role of immune systems in the pathogenesis of post-menopausal OP and included

new bibliographic references on this topic including those suggested ([93],[94],[95],[96],[98],[99]). We inserted the text reported below.

Interestingly, estrogens play an important role in modulating immune responses by controlling the functions of T cells and macrophages with essential implications in bone metabolism [94]. Estrogen deficiency contributes to increasing inflammatory cytokines leading to increased osteoclastogenesis [87][95]. TNF-α, produced by bone marrow T lymphocytes, is among the main cytokines in the estrogen deficiency-induced bone loss [87]. In particular, it enhances OCs formation and increases the responsiveness of pre-OCs to RANKL thanks to NFκB and AP-1 signalling pathways. Moreover, it blocks osteoblastogenesis by impairing the function of bone-forming OBs [96]. The microbiota is among the factors activating T cells in sex steroid deficiency–associated bone loss during OP. Increased gut permeability contributes to triggering inflammatory signalling pathways [97]. Beyond T cells, B cells and macrophages also contribute to the production of osteoclastogenic cytokines [96][98]. Recently, the relationship between macrophage polarisation and estrogens is gaining considerable attention among scientists. Several studies showed that estrogens promote M2 macrophages (with a wound-healing phenotype) and inhibit M1 subset (with a pro-inflammatory phenotype). Estrogens deficiency is responsible for an increased M1/M2 macrophage ratio, leading to increased production of osteoclastogenic factors. Estrogen-dependent effects on macrophage subsets may be a potential target for pharmacological approaches in post-menopausal OP [99].

You can find all these details on Page 7 from Line 277 to 279 and on Page 8 from Line 297 to 310.

  1. About monoclonal antibodies towards osteoporosis the following reference should be added: Expert Opin Biol Ther. 2018 Feb;18(2):149-157. Furthermore, as anabolic agent, the use of romosozumab should be reported.

-) We thank the reviewer for his/her suggestion. We inserted the reference suggested by the reviewer ([124] and [125]) and reported romosozumab as an anabolic agent (Page 9; Lines, 382-383).
